# Antiretroviral therapy retention, adherence, and clinical outcomes among postpartum women with HIV in Nigeria

Clara M. Young[1][¤], Charlotte A. Chang[2], Atiene S. Sagay[3], Godwin Imade[3], Olabanjo O. Ogunsola[4], Prosper Okonkwo[4], Phyllis J. Kanki[2]*

**1** College of Public Health, The University of Iowa, Iowa City, Iowa, United States of America, **2** Department of Immunology and Infectious Diseases, Harvard T.H. Chan School of Public Health, Boston, Massachusetts, United States of America, **3** Jos University Teaching Hospital, University of Jos, Jos, Nigeria, **4** APIN Public Health Initiatives, Abuja, Nigeria

¤ Current address: University of California San Diego, San Diego, California, United States of America
* pkanki@hsph.harvard.edu

**Data Availability Statement:** Data cannot be shared publicly because of confidentiality and ethical restrictions associated with patient data collected in a clinical care program. The data that

## Abstract

While research involving pregnant women with HIV has largely focused on the antepartum and intrapartum periods, few studies in Nigeria have examined the clinical outcomes of these women postpartum. This study aimed to evaluate antiretroviral therapy retention, adherence, and viral suppression among postpartum women in Nigeria. This retrospective clinical data analysis included women with a delivery record at the antenatal HIV clinic at Jos University Teaching Hospital between 2013 and 2017. Descriptive statistics quantified proportions retained, adherent ($\geq$95% medication possession ratio), and virally suppressed up to 24 months postpartum. Among 1535 included women, 1497 met the triple antiretroviral therapy eligibility criteria. At 24 months, 1342 (89.6%) women remained in care, 51 (3.4%) reported transferring, and 104 (7.0%) were lost to follow-up. The proportion of patients with $\geq$95% medication possession ratio decreased from 79.0% to 69.1% over the 24 months. Viral suppression among those with results was 88.7% at 24 months, but <62% of those retained had viral load results at each time point. In multiple logistic regression, predictors of loss to follow-up included having a more recent HIV diagnosis, higher gravidity, fewer antenatal care visits, and a non-hospital delivery. Predictors of viral non-suppression included poorer adherence, unsuppressed/missing baseline viral load, lower baseline CD4+ T-cell count, and higher gravidity. Loss to follow-up rates were lower and antiretroviral therapy adherence rates similar among postpartum women at our study hospital compared with other sub-Saharan countries. Longer follow-up time and inclusion of multiple facilities for a nationally representative sample would be beneficial in future studies.

## Introduction

Since the human immunodeficiency virus (HIV) epidemic's peak in 1995, new infections have decreased by 59% to 1.3 million in 2022 [1]. This wane can be widely attributed to the

support the findings of this study are available on request from the APIN Public Health Initiatives Data Management Committee (contact via dmc@apin. org.ng) for researchers who meet the criteria for access to confidential data.

**Funding:** The author(s) received no specific funding for this work.

**Competing interests:** The authors have declared that no competing interests exist.

accessibility and evolution of antiretroviral therapy (ART), which has positively shifted patient prognosis and decreased transmission risk. Nonetheless, HIV persists as a leading cause of death in low-income countries–remaining a global health priority [2].

Disproportionately affected by the disease, sub-Saharan Africa constituted 60% of global HIV infections in 2020 [3]. Within the central-west subregion, Nigeria bears the greatest disease burden, ranking fourth in the world [4]. The prioritization of HIV programming in the country has resulted in attenuated incidence and HIV-related morbidity and mortality; however challenges persist [5]. New infections continue to be fueled through mother-to-child transmission (MTCT) during gestation, delivery, and breastfeeding, with 21,000 newly infected children in Nigeria in 2020 [3]. ART coverage among pregnant and breastfeeding women living with HIV in Nigeria was estimated at 44%, with a 25% vertical transmission rate in 2020 [3].

While MTCT research has historically centered on ART uptake and viral suppression among pregnant women living with HIV through delivery and diagnostic outcomes among their neonates after delivery, less coverage has been given to ART continuation and clinical outcomes among these women postpartum [6]. Postnatal continuation of maternal ART can reduce MTCT rates to less than 2% in resource-limited countries [7–9]. However, ART adherence declines dramatically in mothers with HIV up to 18 months postpartum [10], and new mothers have increased risks of becoming lost to follow-up (LTFU) in HIV care and having viremia, increasing risk of transmission to infants [11, 12]. Suboptimal adherence or discontinuation of ART endangers maternal health, even increasing the odds of death [13, 14]; with mothers often the primary familial caregivers, poor maternal health, in turn, endangers their children's health.

To achieve the United Nations Programme on HIV/AIDS (UNAIDS) 95-95-95 goals for the year 2030—that 95% of those living with HIV know their status, have ART, and be virally suppressed—it is crucial to assess ART retention, adherence, and viral suppression among all key populations, including postnatal women [15, 16]. We retrospectively measured postpartum retention, ART adherence, and viral suppression in an HIV care program in Nigeria. We additionally identified demographic and clinical risk factors for postpartum LTFU and unsuppressed viral load.

## Materials and methods

### Patient population

The study population included 1535 pregnant women living with HIV-1, ≥18 years of age, attending the antenatal HIV clinic at Jos University Teaching Hospital (JUTH), north-central Nigeria, with a recorded delivery between 2013 and 2017. Demographic and clinical data were extracted from the clinical databases at enrollment in the HIV program, ART initiation, antenatal booking, delivery, and up to 24 months after delivery, with data censored on 31 December 2019. The data were collected for routine clinical management as part of the APIN Public Health Initiatives HIV program at JUTH, supported by the President's Emergency Plan for AIDS Relief (PEPFAR).

Per routine care, patients initiating ART in the adult HIV clinic at JUTH had scheduled medical examinations with CD4+ T-cell counts at baseline and every six months thereafter, and viral load enumeration at 6 months, 12 months, and every 12 months thereafter (with enhanced adherence counseling and additional viral load monitoring if viremic). If a patient with HIV became pregnant or a pregnant woman newly tested HIV-positive, they were enrolled in the antenatal HIV clinic at JUTH. Pregnant women newly initiating treatment received an additional viral load test at initiation. After delivery, they transferred back to the

general adult HIV clinic for routine HIV care and returned to the adult monitoring schedule. If a woman receiving antenatal HIV care at JUTH delivered outside of JUTH, their delivery information was recorded when they returned to the clinic with their newborn. All clinical, laboratory, and pharmacy data collected in medical record forms at the adult and antenatal HIV clinics were routinely entered into electronic databases.

ART eligibility conformed to the Nigerian National Guidelines for HIV Prevention and Care, which were revised over the study years [17–19]. Between 2013 and 2014, all adults with WHO clinical stage III or IV or CD4+ T-cell counts less than 350 cells/mm$^3$ were triple ART-eligible. Ineligible pregnant women received zidovudine and lamivudine during pregnancy with single-dose nevirapine at delivery as prophylaxis. In 2014, Nigerian ART eligibility guidelines were revised to include adults with CD4+ T-cell counts less than 500 cells/mm$^3$; to allow for program implementation delays, we considered women meeting these criteria ART-eligible starting 1 January 2015. In 2016, Nigeria expanded eligibility to all pregnant women with HIV per the WHO Option B+ guideline; to allow time for implementation, we considered all women ART-eligible by 1 January 2017.

This project was approved by the APIN Public Health Initiatives and Harvard T.H. Chan School of Public Health Institutional Review Boards. Data used were from patients who provided written informed consent for the use of their data for secondary research at the time of enrollment in the HIV program. The clinical records, which included identifiable information, were accessed on June 17, 2020 by the data manager, who assembled the dataset and removed all identifiers before statistical analyses.

## Measurement and definitions of outcomes

**ART adherence.**   ART adherence was measured using pharmacy refill data [20]. Medication possession ratios (MPR) were calculated as the percentage of all days supplied with ART over the total days in each 6-month postpartum time interval, and were categorized as <80%, 80%-94.9%, and ≥95%. New patients picked up ART monthly during their first year; if adherent and virally suppressed, they could pick up a two-month supply bimonthly. Patients were excluded from the adherence analysis if they were receiving ART prophylactically. Patients who became LTFU during the study period were included in the analysis up to the time interval during which their last clinic visit was recorded.

**Loss to follow-up.**   While LTFU definitions vary by program and over the years, overlapping with current definitions of interruption in treatment, we defined LTFU conservatively as a sustained absence of ≥180 days since the last clinical and pharmacy visits, assessed at 24 months postpartum [21]. Patients who were ART-ineligible or reported transferring to alternate clinics during the analyzed period were removed from evaluation of this outcome.

**Unsuppressed viral load.**   Following program cut-offs, an unsuppressed viral load was defined as a viral load measurement of ≥1000 viral copies/mL. Since laboratory tests were not always performed precisely at the study time points, we used the viral load results within the following timeframes that were closest to baseline, month 12, and month 24: 6 months before delivery to 15 days postpartum, 6–18 months postpartum, and 18–30 months postpartum, respectively.

## Independent variables

Baseline age, HIV clinical (prior years of ART, years since HIV diagnosis, delivery year, ART regimen, viral load, CD4+ T-cell count, and ART adherence between 0–6 months postpartum), and antenatal (surviving children, gestational age at antenatal booking, total antenatal visits, delivery site, delivery year, delivery type, term delivery, infant birthweight, and infant

feeding method) factors were evaluated for associations with outcomes, with baseline defined as at or closest to the time of delivery. Other demographic information (marital status, education, and occupation) and previous ART experience were only collected in the ART enrollment record, which was completed whenever the patient initiated ART in the PEPFAR program.

## Statistical analysis

Continuous variables were converted to ordinal by interquartile range (IQR) and clinical categories. Retention, ART adherence, and viral suppression were measured using simple descriptive statistics. Bivariate analyses were performed to identify potential associations between the independent variables and the following outcomes: LTFU and unsuppressed viral load. Exposure variables with a chi-square p-value <0.2 (or Fisher's exact for frequencies ≤5) were evaluated in a multiple logistic regression model and retained in the final model if p ≤0.05, after backwards elimination. To account for significant missing viral load data, missing baseline viral load results were coded as a separate category, and a sensitivity analysis was performed to compare those with versus those missing postpartum viral load results using bivariate analyses and multiple logistic regression as above. Analysis was conducted with SAS Studio 2020.1.2.

## Results

### Patient population characteristics

1535 women with delivery records at the JUTH antenatal HIV clinic were included in the analyses (Table 1). The median age at delivery was 33 years (IQR: 29–36). The majority of women were married (66.8%), achieved primary or secondary education (61.7%), held non-income generating occupations (46.2%), and lacked previous ART experience (88.7%) at the time of ART enrollment in the PEPFAR HIV program. Most women were ART-eligible (90.6%) at the time of delivery. The median time since HIV diagnosis was 6.3 years (IQR: 3.3–8.4) and the median duration on ART was 5.6 years (IQR:2.9–8.1). Most patients (91.1%) were receiving an ART regimen without a protease inhibitor at delivery. At delivery, the majority of patients (87.2%) had ≥200 CD4+ T-cells/mm$^3$ and 51.9% of patients were virally suppressed, with 38.6% of patients missing a viral load result.

### ART retention

Among 1497 ART-eligible women at delivery, 1342 (89.6%) were retained in care at 24 months postpartum (Fig 1). Cumulatively, 51 (3.4%) women reported transferring to another clinic, and 104 (7.0%) were LTFU.

### ART adherence over time

Among ART-eligible women retained in each period, mean MPR over time was 95.5% (95% CI 95.0%–96.1%) between 0–6 months, 93.9% (95% CI 93.1%–94.6%) between 6–12 months, 92.0% (95% CI 91.0%–92.9%) between 12–18 months, and 91.0% (95% CI 90.0%–91.9%) between 18–24 months postpartum. The proportion of postpartum patients with ≥95% MPR decreased from 79.0% to 69.1% while the proportion of patients with <80% MPR increased from 7.4% to 15.7% over the study period (Fig 2).

### Viral load suppression

Among all 1497 ART-eligible patients, 926 (61.9%) had a baseline viral load result, 688 (50.0%) had a viral load result at 12 months postpartum, and 858 (57.3%) had a viral load result at 24

**Table 1. Baseline characteristics of study population.**

| Demographic | | Number | % |
|---|---|---|---|
| Age at Delivery | Missing Data | 1 | . |
| | ≤29 years | 395 | 25.75 |
| | 30–33 years | 458 | 29.86 |
| | 34–36 years | 326 | 21.25 |
| | ≥37 years | 355 | 23.14 |
| Marital Status[a] | Missing Data | 39 | . |
| | Single/Separated/Divorced | 496 | 33.16 |
| | Married | 1000 | 66.84 |
| Education Status[a] | Missing Data | 42 | . |
| | No Formal | 130 | 8.71 |
| | Primary/Secondary | 921 | 61.69 |
| | Tertiary | 442 | 29.6 |
| Occupation Status[a] | Missing Data | 45 | . |
| | Non-income Generating | 688 | 46.17 |
| | Professional/Manager | 356 | 23.89 |
| | Labor/Service | 446 | 29.93 |
| **Clinical HIV/ART**[b] | | **Number** | **%** |
| Previous ART Experience[a] | Missing Data | 39 | . |
| | ART Naive | 1327 | 88.7 |
| | ART Experienced | 169 | 11.3 |
| Time since HIV Diagnosis | Missing Data | 55 | . |
| | Diagnosis during Pregnancy/Delivery | 144 | 9.73 |
| | ≤3 years Prepartum | 198 | 13.38 |
| | 3.1–6 years Prepartum | 367 | 24.8 |
| | 6.1–8 years Prepartum | 337 | 22.77 |
| | >8 years Prepartum | 434 | 29.32 |
| Duration on ART prior to Delivery | Missing Data | 95 | . |
| | <4 years | 487 | 33.82 |
| | 4–8 years | 586 | 40.69 |
| | >8 years | 367 | 25.49 |
| Drug Regimen at Delivery | Missing Data | 30 | . |
| | Regimens without a Protease Inhibitor | 1371 | 91.1 |
| | Regimens with a Protease Inhibitor | 134 | 8.9 |
| Viral Load at Delivery | Missing Data | 592 | 38.57 |
| | Suppressed (<200 copies/mL) | 706 | 45.99 |
| | Suppressed (200–999 copies/mL) | 91 | 5.93 |
| | Unsuppressed (≥1000 copies/mL) | 146 | 9.51 |
| CD4 Cell Count at Delivery | Missing Data | 147 | . |
| | <200 cells/mm$^3$ | 178 | 12.82 |
| | 200–349 cells/mm$^3$ | 402 | 28.96 |
| | 350–500 cells/mm$^3$ | 421 | 30.33 |
| | >500 cells/mm$^3$ | 387 | 27.88 |
| **Antenatal** | | **Number** | **%** |
| Plurality | Missing Data | 175 | . |
| | 1 | 1338 | 98.38 |
| | 2–3 | 22 | 1.62 |
| Gravidity | Missing Data | 95 | . |

(*Continued*)

**Table 1.** (Continued)

| Demographic | | Number | % |
|---|---|---|---|
| | 1 | 96 | 6.67 |
| | 2–3 | 523 | 36.32 |
| | ≥4 | 821 | 57.01 |
| Previous Live Births | Missing Data | 147 | . |
| | 0 | 198 | 14.27 |
| | 1 | 301 | 21.69 |
| | ≥2 | 889 | 64.05 |
| Surviving Children | Missing Data | 149 | . |
| | 0 | 239 | 17.24 |
| | 1–2 | 725 | 52.31 |
| | >2 | 422 | 30.45 |
| Previous Abortion | Missing Data | 198 | . |
| | 0 | 797 | 59.61 |
| | ≥1 | 540 | 40.39 |
| Trimester at First Antenatal Care Visit | Missing Data | 11 | . |
| | 1st (≤12 weeks) | 113 | 7.41 |
| | 2nd (13–26 weeks) | 989 | 64.9 |
| | 3rd (≥27 weeks) | 422 | 27.69 |
| Total Antenatal Care Visits | 1–2 | 329 | 21.43 |
| | 3–4 | 606 | 39.48 |
| | >4 | 600 | 39.09 |
| **Delivery** | | **Number** | **%** |
| Year of Delivery | 2013–2014 | 745 | 48.53 |
| | 2015–2017 | 790 | 51.47 |
| Delivery Site | Missing Data | 14 | . |
| | Jos University Teaching Hospital | 276 | 18.15 |
| | Other Clinic/Home/Road | 1245 | 81.85 |
| Delivery Type | Missing Data | 108 | . |
| | Vaginal/Assisted | 1086 | 76.1 |
| | Emergency/Elective C-Section | 341 | 23.9 |
| Gestational Age at Delivery | Missing Data | 127 | . |
| | Pre-term (<37 weeks) | 64 | 4.55 |
| | Full-term (≥37 weeks) | 1344 | 95.45 |
| Infant Birthweight | Missing Data | 132 | . |
| | Low (≤2.5 kg) | 375 | 26.73 |
| | Normal/High (>2.5 kg) | 1028 | 73.27 |
| Infant Feed Method at Delivery | Missing Data | 33 | . |
| | Exclusive Breast Feeding | 1345 | 89.55 |
| | Breast Milk Substitute Supplement | 157 | 10.45 |

[a]Denotes variables collected from the ART enrollment record, which was completed when the patient initiated ART in the APIN PEPFAR program. All other variables collected at antenatal booking or at delivery, as indicated.

[b]ART, antiretroviral therapy.

months postpartum. Mean viral loads among all recorded results were 26,993 viral copies/mL (95% CI 12,551–41,435) at delivery, 9206 viral copies/mL (95% CI 2107–16,305) at 12 months, and 5796 viral copies/mL (95% CI 3256–8336) at 24 months. Among those with VL results, the

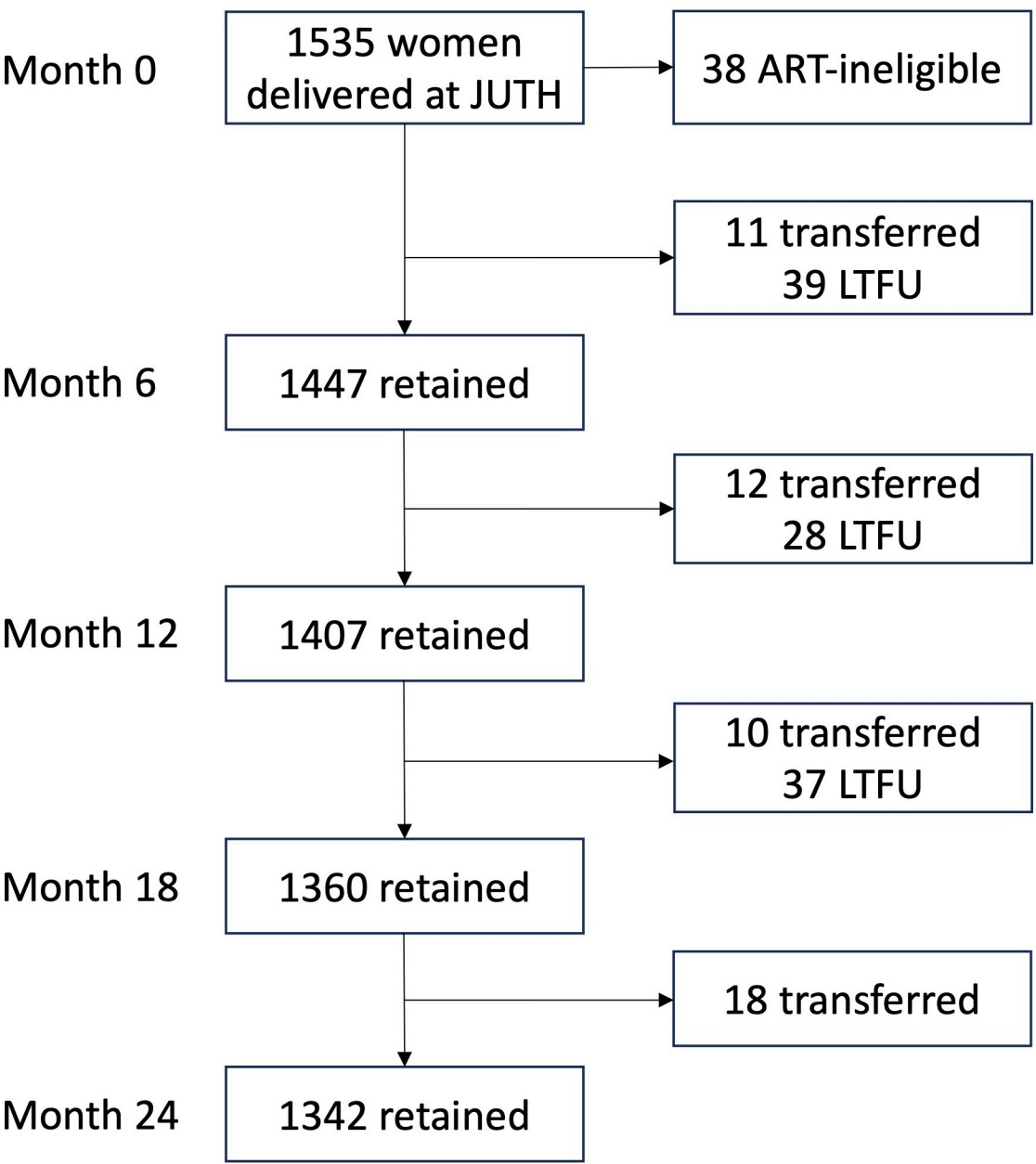

**Fig 1. Study population flowchart.** Representation of study flow showing numbers of included individuals retained and not retained at each 6-month timepoint. Abbreviations: JUTH, Jos University Teaching Hospital; LTFU, lost to follow-up.

proportion of patients with a suppressed viral load was 84.9% (786/926) at delivery, 85.8% (590/688) at 12 months, and 88.7% (761/858) at 24 months.

### Risk factors for LTFU

In chi-square bivariate analysis (Table 2), LTFU was potentially associated with the following variables: maternal age, previous ART experience, years since HIV diagnosis, duration on ART, viral load at delivery, CD4+ T-cell count at delivery, total antenatal care visits, abortion history, gravidity, total surviving children, and delivery site.

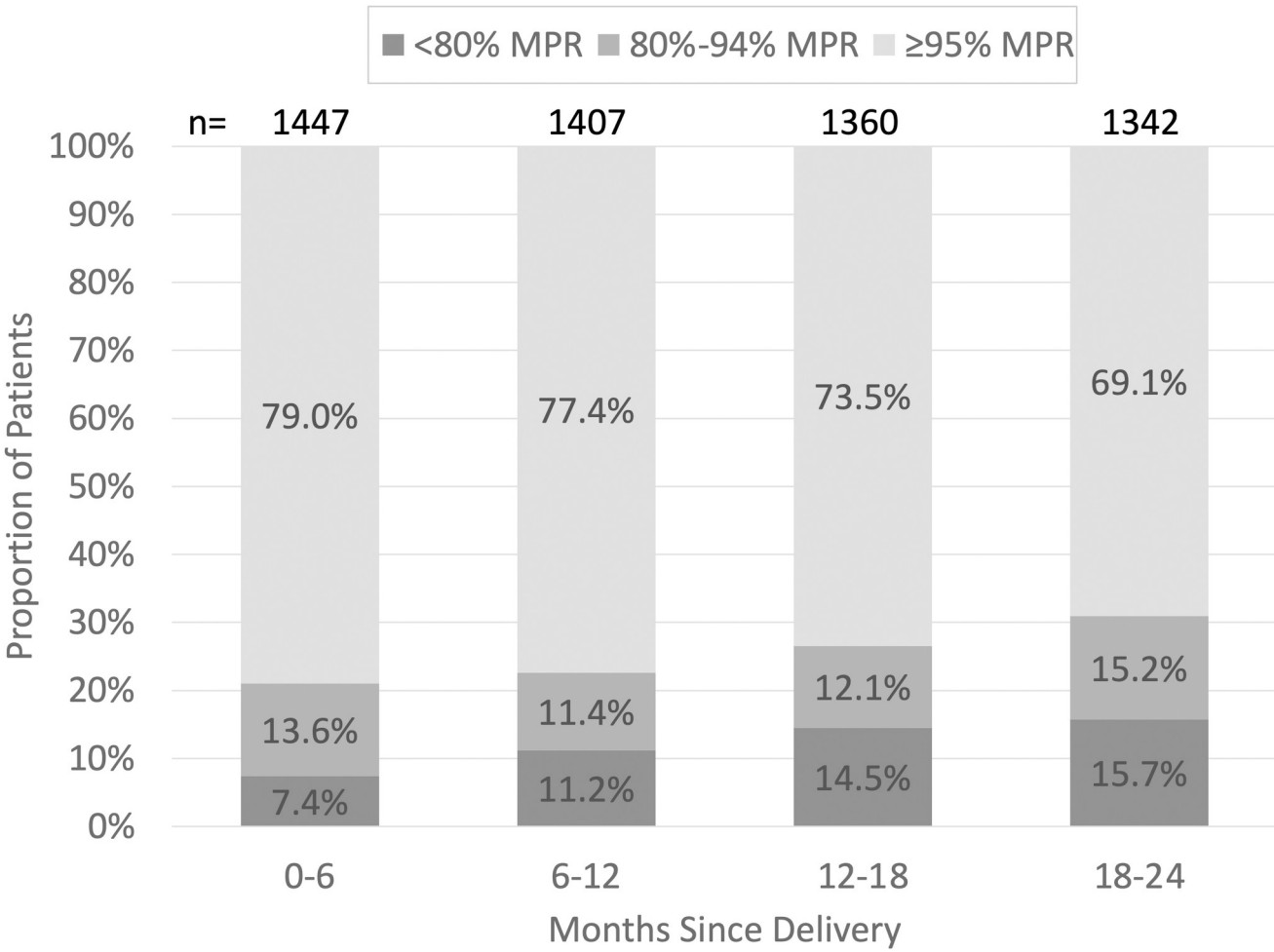

**Fig 2. Medication possession ratio.** The proportion of patients with <80%, 80%-94.9%, and ≥95% MPR between 0–6, 6–12, 12–18, and 18–24 months after delivery among ART-eligible patients retained in each period. Abbreviations: MPR, medication possession ratio; n, number of patients retained.

1303 patients were retained in the final multiple logistic regression model for risk factors associated with LTFU (Fig 3). A longer time since HIV diagnosis (3.1–6 years, aOR = 0.421, 95% CI 0.202–0.876; 6.1–8 years, aOR = 0.347, 95% CI 0.161–0.745; >8 years, aOR = 0.231, 95% CI 0.106–0.502) and having attended >4 antenatal care visits (aOR = 0.312, 95% CI 0.171–0.568) significantly decreased risk of becoming LTFU. Alternatively, having gravidity of ≥4 pregnancies (aOR = 3.733, 95% CI 1.095–12.73) and delivering outside of JUTH (aOR = 2.752, 95% CI 1.166–6.497) significantly increased risk for becoming LTFU.

### Risk factors for unsuppressed viral load

In chi-square bivariate analysis (Table 3), unsuppressed viral load was potentially associated with the following variables: marital status, education level, occupation, previous ART experience, delivery site, total antenatal care visits, gestational age at antenatal booking, previous live births, surviving children, abortion history, drug regimen, and infant feeding method.

The final multiple logistic regression model for unsuppressed viral load retained 866 patients (Fig 4). Having ≥95% ART adherence (aOR = 0.51, 95% CI 0.319–0.816) and higher CD4+ T-cell counts at delivery (200–349 cells/mm³, aOR = 0.453, 95%CI 0.252–0.814; 350–

**Table 2. Bivariate analysis of loss to follow up.**

| | Retained | | Loss to Follow Up | | Total | chi-square test |
|---|---|---|---|---|---|---|
| | Number | % | Number | % | Number | p-value |
| **Demographic** | | | | | | |
| Age at Delivery | | | | | | |
| ≤29 years | 311 | 88.6% | 40 | 11.4% | 351 | 0.0044 |
| 30–33 years | 412 | 93.4% | 29 | 6.6% | 441 | |
| 34–36 years | 298 | 95.2% | 15 | 4.8% | 313 | |
| ≥37 years | 320 | 94.1% | 20 | 5.9% | 340 | |
| Marital Status[a] | | | | | | |
| Single/Separated/Divorced | 441 | 93.2% | 32 | 6.8% | 473 | 0.6998 |
| Married | 873 | 92.7% | 69 | 7.3% | 942 | |
| Education Status[a] | | | | | | |
| No Formal | 107 | 90.7% | 11 | 9.3% | 118 | 0.3037 |
| Primary/Secondary | 808 | 92.4% | 66 | 7.6% | 874 | |
| Tertiary | 397 | 94.3% | 24 | 5.7% | 421 | |
| Occupation Status[a] | | | | | | |
| Non-income Generating | 607 | 93.0% | 46 | 7.0% | 653 | 0.3927 |
| Professional/Manager | 314 | 94.3% | 19 | 5.7% | 333 | |
| Labor/Service | 388 | 91.7% | 35 | 8.3% | 423 | |
| **Clinical HIV/ART[b]** | | | | | | |
| Previous ART Experience[a] | | | | | | |
| ART Naive | 1167 | 92.5% | 94 | 7.5% | 1261 | 0.1856 |
| ART Experienced | 147 | 95.5% | 7 | 4.5% | 154 | |
| Time since HIV Diagnosis | | | | | | |
| Diagnosis during Pregnancy/Delivery | 111 | 88.1% | 15 | 11.9% | 126 | 0.0215 |
| ≤3 years Prepartum | 163 | 90.1% | 18 | 9.9% | 181 | |
| 3.1–6 years Prepartum | 324 | 92.3% | 27 | 7.7% | 351 | |
| 6.1–8 years Prepartum | 303 | 93.8% | 20 | 6.2% | 323 | |
| >8 years Prepartum | 401 | 95.5% | 19 | 4.5% | 420 | |
| Duration on ART prior to Delivery | | | | | | |
| <4 years | 406 | 90.0% | 45 | 10.0% | 451 | 0.0052 |
| 4–8 years | 532 | 94.0% | 34 | 6.0% | 566 | |
| >8 years | 340 | 95.5% | 16 | 4.5% | 356 | |
| Drug Regimen at Delivery | | | | | | |
| Regimens without a Protease Inhibitor | 1226 | 93.7% | 82 | 6.3% | 1308 | 0.4657 |
| Regimens with a Protease Inhibitor | 116 | 92.1% | 10 | 7.9% | 126 | |
| Viral Load at Delivery | | | | | | |
| Suppressed (<1000 copies/mL) | 724 | 95.1% | 37 | 4.9% | 761 | 0.0011 |
| Unsuppressed (≥1000 copies/mL) | 121 | 91.7% | 11 | 8.3% | 132 | |
| Missing Data | 497 | 89.9% | 56 | 10.1% | 553 | |
| CD4 Cell Count at Delivery | | | | | | |
| <200 cells/mm$^3$ | 153 | 89.0% | 19 | 11.0% | 172 | 0.012 |
| 200–349 cells/mm$^3$ | 359 | 93.2% | 26 | 6.8% | 385 | |
| 350–500 cells/mm$^3$ | 386 | 95.8% | 17 | 4.2% | 403 | |
| >500 cells/mm$^3$ | 342 | 95.0% | 18 | 5.0% | 360 | |
| **Antenatal** | | | | | | |
| Plurality | | | | | | |
| 1 | 1169 | 92.7% | 92 | 7.3% | 1261 | 1[c] |

*(Continued)*

**Table 2.** (*Continued*)

| | Retained | | Loss to Follow Up | | Total | chi-square test |
|---|---|---|---|---|---|---|
| | Number | % | Number | % | Number | p-value |
| 2–3 | 19 | 95.0% | 1 | 5.0% | 20 | |
| Gravidity | | | | | | |
| 1 | 83 | 96.5% | 3 | 3.5% | 86 | 0.0527[c] |
| 2–3 | 457 | 94.4% | 27 | 5.6% | 484 | |
| ≥4 | 717 | 91.3% | 68 | 8.7% | 785 | |
| Previous Live Births | | | | | | |
| 0 | 166 | 92.2% | 14 | 7.8% | 180 | 0.2881 |
| 1 | 267 | 95.0% | 14 | 5.0% | 281 | |
| ≥2 | 779 | 92.3% | 65 | 7.7% | 844 | |
| Surviving Children | | | | | | |
| 0 | 201 | 93.5% | 14 | 6.5% | 215 | 0.1455 |
| 1–2 | 645 | 93.9% | 42 | 6.1% | 687 | |
| >2 | 364 | 90.8% | 37 | 9.2% | 401 | |
| Previous Abortion | | | | | | |
| 0 | 696 | 93.9% | 45 | 6.1% | 741 | 0.0764 |
| ≥1 | 473 | 91.3% | 45 | 8.7% | 518 | |
| Trimester at First Antenatal Care Visit | | | | | | |
| 1st (≤12 weeks) | 99 | 94.3% | 6 | 5.7% | 105 | 0.0705 |
| 2nd (13–26 weeks) | 872 | 93.7% | 59 | 6.3% | 931 | |
| 3rd (≥27 weeks) | 360 | 90.2% | 39 | 9.8% | 399 | |
| Total Antenatal Care Visits | | | | | | |
| 1–2 | 273 | 88.3% | 36 | 11.7% | 309 | < .0001 |
| 3–4 | 529 | 92.0% | 46 | 8.0% | 575 | |
| >4 | 540 | 96.1% | 22 | 3.9% | 562 | |
| **Delivery** | | | | | | |
| Year of Delivery | | | | | | |
| 2013–2014 | 637 | 92.6% | 51 | 7.4% | 688 | 0.7571 |
| 2015–2017 | 705 | 93.0% | 53 | 7.0% | 758 | |
| Delivery Site | | | | | | |
| Jos University Teaching Hospital | 254 | 96.6% | 9 | 3.4% | 263 | 0.0122 |
| Other Clinic/Home/Road | 1079 | 92.2% | 91 | 7.8% | 1170 | |
| Delivery Type | | | | | | |
| Vaginal/Assisted | 943 | 92.1% | 81 | 7.9% | 1024 | 0.0855 |
| Emergency/Elective C-Section | 301 | 95.0% | 16 | 5.0% | 317 | |
| Gestational Age at Delivery | | | | | | |
| Pre-term (<37 weeks) | 57 | 95.0% | 3 | 5.0% | 60 | 0.7952[c] |
| Full-term (≥37 weeks) | 1174 | 92.8% | 91 | 7.2% | 1265 | |
| Infant Birthweight | | | | | | |
| Low (≤2.5 kg) | 320 | 90.9% | 32 | 9.1% | 352 | 0.0791 |
| Normal/High (>2.5 kg) | 908 | 93.7% | 61 | 6.3% | 969 | |
| Infant Feed Method at Delivery | | | | | | |
| Exclusive Breast Feeding | 1180 | 93.1% | 87 | 6.9% | 1267 | 0.7975 |

(*Continued*)

**Table 2.** (Continued)

| | Retained | | Loss to Follow Up | | Total | chi-square test |
|---|---|---|---|---|---|---|
| | **Number** | **%** | **Number** | **%** | **Number** | **p-value** |
| Breast Milk Substitute Supplement | 137 | 92.6% | 11 | 7.4% | 148 | |

[a]Denotes variables collected from the ART enrollment record, which was completed when the patient initiated ART in the APIN PEPFAR program. All other variables collected at antenatal booking or at delivery, as indicated.

[b]ART, antiretroviral therapy.

[c]Fisher's exact test p-values reported when contingency table observations were less than or equal to five.

500 cells/mm$^3$, aOR = 0.284, 95% CI 0.152–0.528; >500 cells/mm$^3$, aOR = 0.235, 95% CI 0.121–0.456) were significantly protective against experiencing unsuppressed viral load post-partum. Significant risk factors for this adverse outcome were having an unsuppressed or missing viral load at delivery (aOR = 13.128, 95% CI 7.147–24.115 and aOR = 2.402, 95% CI 1.534–3.761, respectively) and having gravidity of ≥4 pregnancies (aOR = 2.716, 95% CI 1.052–7.015).

Our sensitivity analysis compared 422 women missing vs. 1024 women not missing post-partum viral load data (Table 4). Maternal age, marital status, time since HIV diagnosis, ART adherence, ART duration, viral load at delivery, CD4+ T-cell count at delivery, delivery year, and delivery type were potentially associated with missing data in bivariate analyses. The final multiple logistic regression model indicated that missing postpartum viral load data had significant inverse associations with having been on ART >8 years, ≥95% MPR, and delivery year after 2014.

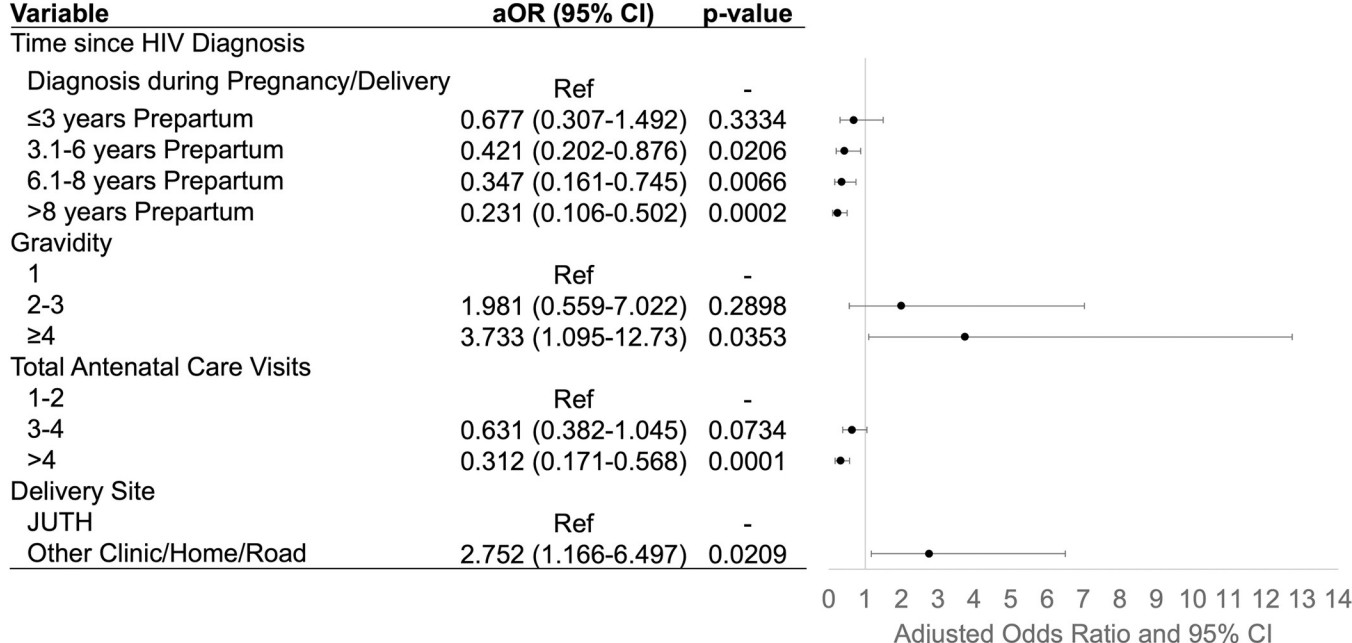

**Fig 3. Risk factors for postpartum loss to follow-up.** Final multiple logistic regression model shows significant risk factors for women becoming lost to follow-up from the Jos University Teaching Hospital HIV clinic after delivery up to 24 months postpartum. Abbreviations: aOR, adjusted odds ratio; CI, confidence interval; ref, reference group.

**Table 3. Bivariate analysis of unsuppressed viral load.**

| | Suppressed Viral Load | | Unsuppressed Viral Load | | Total | chi-square test |
|---|---|---|---|---|---|---|
| | Number | % | Number | % | Number | p-value |
| **Demographic** | | | | | | |
| Age at Delivery | | | | | | |
| ≤29 years | 189 | 85.5% | 32 | 14.5% | 221 | 0.4123 |
| 30–33 years | 263 | 83.2% | 53 | 16.8% | 316 | |
| 34–36 years | 182 | 81.6% | 41 | 18.4% | 223 | |
| ≥37 years | 228 | 86.7% | 35 | 13.3% | 263 | |
| Marital Status[a] | | | | | | |
| Single/Separated/Divorced | 305 | 86.9% | 46 | 13.1% | 351 | 0.0663 |
| Married | 535 | 82.4% | 114 | 17.6% | 649 | |
| Education Status[a] | | | | | | |
| No Formal | 50 | 67.6% | 24 | 32.4% | 74 | 0.0002 |
| Primary/Secondary | 522 | 84.9% | 93 | 15.1% | 615 | |
| Tertiary | 267 | 86.4% | 42 | 13.6% | 309 | |
| Occupation Status[a] | | | | | | |
| Non-income Generating | 370 | 81.5% | 84 | 18.5% | 454 | 0.1074 |
| Professional/Manager | 217 | 87.5% | 31 | 12.5% | 248 | |
| Labor/Service | 249 | 84.7% | 45 | 15.3% | 294 | |
| **Clinical HIV/ART[b]** | | | | | | |
| Previous ART Experience[a] | | | | | | |
| ART Naive | 749 | 84.7% | 135 | 15.3% | 884 | 0.0828 |
| ART Experienced | 91 | 78.4% | 25 | 21.6% | 116 | |
| Time since HIV Diagnosis | | | | | | |
| Diagnosis during Pregnancy/Delivery | 61 | 83.6% | 12 | 16.4% | 73 | 0.3027 |
| ≤3 years Prepartum | 104 | 86.7% | 16 | 13.3% | 120 | |
| 3.1–6 years Prepartum | 199 | 84.7% | 36 | 15.3% | 235 | |
| 6.1–8 years Prepartum | 169 | 79.7% | 43 | 20.3% | 212 | |
| >8 years Prepartum | 305 | 86.2% | 49 | 13.8% | 354 | |
| Duration on ART prior to Delivery | | | | | | |
| <4 years | 243 | 85.3% | 42 | 14.7% | 285 | 0.2335 |
| 4–8 years | 310 | 81.6% | 70 | 18.4% | 380 | |
| >8 years | 264 | 86.0% | 43 | 14.0% | 307 | |
| Drug Regimen at Delivery | | | | | | |
| Regimens without a Protease Inhibitor | 788 | 85.2% | 137 | 14.8% | 925 | 0.0143 |
| Regimens with a Protease Inhibitor | 75 | 75.8% | 24 | 24.2% | 99 | |
| Viral Load at Delivery | | | | | | |
| Suppressed (<1000 copies/mL) | 509 | 91.7% | 46 | 8.3% | 555 | < .0001 |
| Unsuppressed (≥1000 copies/mL) | 30 | 39.0% | 47 | 61.0% | 77 | |
| Missing Data | 324 | 82.7% | 68 | 17.3% | 392 | |
| CD4 Cell Count at Delivery | | | | | | |
| <200 cells/mm$^3$ | 64 | 61.0% | 41 | 39.0% | 105 | < .0001 |
| 200–349 cells/mm$^3$ | 210 | 80.5% | 51 | 19.5% | 261 | |
| 350–500 cells/mm$^3$ | 263 | 88.9% | 33 | 11.1% | 296 | |
| >500 cells/mm$^3$ | 248 | 90.8% | 25 | 9.2% | 273 | |
| Adherence 0–6 Months Postpartum | | | | | | |
| <95% Medication Possession Ratio | 160 | 76.2% | 50 | 23.8% | 210 | 0.0003 |
| ≥95% Medication Possession Ratio | 703 | 86.4% | 111 | 13.6% | 814 | |

*(Continued)*

**Table 3.** (Continued)

| | Suppressed Viral Load | | Unsuppressed Viral Load | | Total | chi-square test |
|---|---|---|---|---|---|---|
| | Number | % | Number | % | Number | p-value |
| **Antenatal** | | | | | | |
| Plurality | | | | | | |
| 1 | 735 | 83.0% | 151 | 17.0% | 886 | 0.4878[c] |
| 2–3 | 14 | 93.3% | 1 | 6.7% | 15 | |
| Gravidity | | | | | | |
| 1 | 59 | 89.4% | 7 | 10.6% | 66 | 0.0751 |
| 2–3 | 295 | 87.0% | 44 | 13.0% | 339 | |
| ≥4 | 450 | 82.1% | 98 | 17.9% | 548 | |
| Previous Live Births | | | | | | |
| 0 | 121 | 90.3% | 13 | 9.7% | 134 | 0.0226 |
| 1 | 172 | 88.2% | 23 | 11.8% | 195 | |
| ≥2 | 486 | 82.4% | 104 | 17.6% | 590 | |
| Surviving Children | | | | | | |
| 0 | 133 | 85.8% | 22 | 14.2% | 155 | 0.0232 |
| 1–2 | 417 | 87.2% | 61 | 12.8% | 478 | |
| >2 | 227 | 79.9% | 57 | 20.1% | 284 | |
| Previous Abortion | | | | | | |
| 0 | 436 | 82.4% | 93 | 17.6% | 529 | 0.0101 |
| ≥1 | 315 | 88.7% | 40 | 11.3% | 355 | |
| Trimester at First Antenatal Care Visit | | | | | | |
| 1st (≤12 weeks) | 67 | 80.7% | 16 | 19.3% | 83 | 0.0513 |
| 2nd (13–26 weeks) | 565 | 86.7% | 87 | 13.3% | 652 | |
| 3rd (≥27 weeks) | 225 | 80.9% | 53 | 19.1% | 278 | |
| Total Antenatal Care Visits | | | | | | |
| 1–2 | 176 | 80.4% | 43 | 19.6% | 219 | 0.1798 |
| 3–4 | 354 | 85.9% | 58 | 14.1% | 412 | |
| >4 | 333 | 84.7% | 60 | 15.3% | 393 | |
| **Delivery** | | | | | | |
| Year of Delivery | | | | | | |
| 2013–2014 | 285 | 84.1% | 54 | 15.9% | 339 | 0.8984 |
| 2015–2017 | 578 | 84.4% | 107 | 15.6% | 685 | |
| Delivery Site | | | | | | |
| Jos University Teaching Hospital | 148 | 80.9% | 35 | 19.1% | 183 | 0.1797 |
| Other Clinic/Home/Road | 707 | 84.9% | 126 | 15.1% | 833 | |
| Delivery Type | | | | | | |
| Vaginal/Assisted | 585 | 83.7% | 114 | 16.3% | 699 | 0.2351 |
| Emergency/Elective C-Section | 206 | 86.9% | 31 | 13.1% | 237 | |
| Gestational Age at Delivery | | | | | | |
| Pre-term (<37 weeks) | 35 | 79.5% | 9 | 20.5% | 44 | 0.3832 |
| Full-term (≥37 weeks) | 739 | 84.5% | 136 | 15.5% | 875 | |
| Infant Birthweight | | | | | | |
| Low (≤2.5 kg) | 193 | 82.8% | 40 | 17.2% | 233 | 0.5975 |
| Normal/High (>2.5 kg) | 580 | 84.3% | 108 | 15.7% | 688 | |
| Infant Feed Method at Delivery | | | | | | |
| Exclusive Breast Feeding | 774 | 85.3% | 133 | 14.7% | 907 | 0.0069 |

*(Continued)*

**Table 3.** (Continued)

| | Suppressed Viral Load | | Unsuppressed Viral Load | | Total | chi-square test |
|---|---|---|---|---|---|---|
| | Number | % | Number | % | Number | p-value |
| Breast Milk Substitute Supplement | 71 | 74.7% | 24 | 25.3% | 95 | |

[a]Denotes variables collected from the ART enrollment record, which was completed when the patient initiated ART in the APIN PEPFAR program. All other variables collected at antenatal booking or at delivery, as indicated.

[b]ART, antiretroviral therapy.

[c]Fisher's exact test p-values reported when contingency table observations were less than or equal to five.

## Discussion

To our knowledge, this retrospective analysis is the first to quantify both ART adherence and viral suppression up to 24 months postpartum and identify risk factors for LTFU and unsuppressed viral load in postpartum women with HIV in Nigeria.

In this study, 69.1% of patients had ≥95% MPR by 24 months postpartum–a proportion comparable to numbers observed in other sub-Saharan countries. Studies in Malawi using prescription pick-up data and South Africa and Zambia using self-reported adherence found 67%, 63.9%, and 70.5% of postpartum women with optimal adherence, respectively [22–24]. Definitions for optimal adherence varied slightly between 90% in the Malawi study and 100% in the South Africa and Zambia studies. We found a lower proportion of adherent postpartum women than a study in Abuja, Nigeria which found 82.9% adherent women using pill count [25].

Lower adherence rates in postpartum women compared with pregnant women with HIV have been documented [26]. We found a downward trend in ART adherence over the 24 months postpartum, from 79.0% of women with ≥95% MPR between months 0–6 to 69.1% of

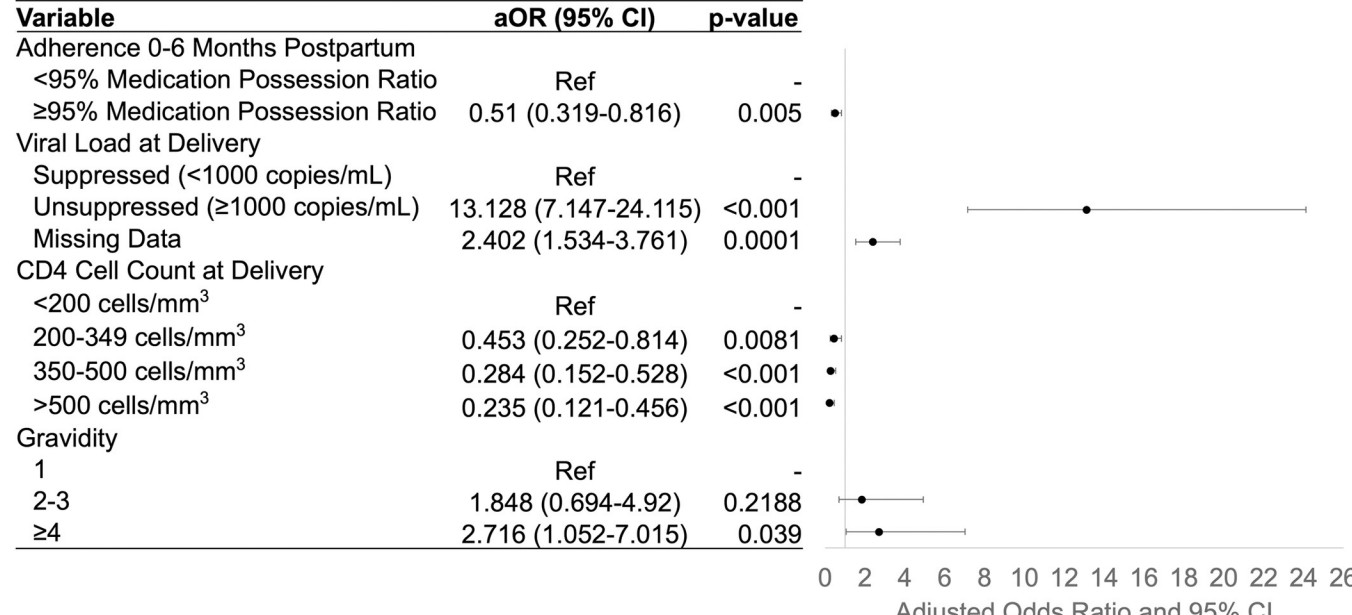

**Fig 4. Risk factors for postpartum unsuppressed viral load.** Final multiple logistic regression model shows significant risk factors for viral load non-suppression after delivery up to 24 months postpartum. Abbreviations: aOR, adjusted odds ratio; CI, confidence interval; ref, reference group.

**Table 4. Sensitivity analysis–Patients with postpartum viral load data vs. patients missing data.**

| | Viral Load Recorded | Viral Load Missing | chi-square test | Multiple Logistic Regression | |
|---|---|---|---|---|---|
| | Number (%) | Number (%) | p-value | aOR[a] (95% CI[b]) | p-value |
| **Demographic** | | | | | |
| Age at Delivery | | | | | |
| ≤29 years | 221 (63.0%) | 130 (37.0%) | 0.0005 | | |
| 30–33 years | 316 (71.7%) | 125 (28.3%) | | | |
| 34–36 years | 223 (71.2%) | 90 (28.8%) | | | |
| ≥37 years | 263 (77.4%) | 77 (22.6%) | | | |
| Marital Status[c] | | | | | |
| Single/Separated/Divorced | 351 (74.2%) | 122 (25.8%) | 0.0384 | | |
| Married | 649 (68.9%) | 293 (31.1%) | | | |
| Education Status[c] | | | | | |
| No Formal | 74 (62.7%) | 44 (37.3%) | 0.0762 | | |
| Primary/Secondary | 615 (70.4%) | 259 (29.6%) | | | |
| Tertiary | 309 (73.4%) | 112 (26.6%) | | | |
| Occupation Status[c] | | | | | |
| Non-income Generating | 454 (69.5%) | 199 (30.5%) | 0.2213 | | |
| Professional/Manager | 248 (74.5%) | 85 (25.5%) | | | |
| Labor/Service | 294 (69.5%) | 129 (30.5%) | | | |
| **Clinical HIV/ART[d]** | | | | | |
| Previous ART Experience[c] | | | | | |
| ART Naive | 884 (70.1%) | 377 (29.9%) | 0.1791 | | |
| ART Experienced | 116 (75.3%) | 38 (24.7%) | | | |
| Time since HIV Diagnosis | | | | | |
| Diagnosis during Pregnancy/Delivery | 73 (57.9%) | 53 (42.1%) | < .0001 | | |
| ≤3 years Prepartum | 120 (66.3%) | 61 (33.7%) | | | |
| 3.1–6 years Prepartum | 235 (67.0%) | 116 (33.0%) | | | |
| 6.1–8 years Prepartum | 212 (65.6%) | 111 (34.4%) | | | |
| >8 years Prepartum | 354 (84.3%) | 66 (15.7%) | | | |
| Duration on ART prior to Delivery | | | | | |
| <4 years | 285 (63.2%) | 166 (36.8%) | < .0001 | Ref[e] | Ref |
| 4–8 years | 380 (67.1%) | 186 (32.9%) | | 0.88 (0.66–1.18) | 0.3935 |
| >8 years | 307 (86.2%) | 49 (13.8%) | | 0.47 (0.32–0.70) | 0.0002 |
| Drug Regimen at Delivery | | | | | |
| Regimens without a Protease Inhibitor | 925 (70.7%) | 383 (29.3%) | 0.0624 | | |
| Regimens with a Protease Inhibitor | 99 (78.6%) | 27 (21.4%) | | | |
| Viral Load at Delivery | | | | | |
| Suppressed (<1000 copies/mL) | 555 (72.9%) | 206 (27.1%) | 0.0030 | | |
| Unsuppressed (≥1000 copies/mL) | 77 (58.3%) | 55 (41.7%) | | | |
| Missing Data | 392 (70.9%) | 161 (29.1%) | | | |
| CD4 Cell Count at Delivery | | | | | |
| <200 cells/mm$^3$ | 105 (61.0%) | 67 (39.0%) | 0.0015 | | |
| 200–349 cells/mm$^3$ | 261 (67.8%) | 124 (32.2%) | | | |
| 350–500 cells/mm$^3$ | 296 (73.4%) | 107 (26.6%) | | | |
| >500 cells/mm$^3$ | 273 (75.8%) | 87 (24.2%) | | | |
| Adherence 0–6 Months Postpartum | | | | | |
| <95% Medication Possession Ratio | 210 (64.2%) | 117 (35.8%) | 0.0023 | Ref | Ref |
| ≥95% Medication Possession Ratio | 814 (72.9%) | 302 (27.1%) | | 0.40 (0.28–0.56) | <0.001 |

(*Continued*)

**Table 4.** (*Continued*)

|  | Viral Load Recorded | Viral Load Missing | chi-square test | Multiple Logistic Regression | |
|---|---|---|---|---|---|
|  | Number (%) | Number (%) | p-value | aOR[a] (95% CI[b]) | p-value |
| **Antenatal** |  |  |  |  |  |
| Plurality |  |  |  |  |  |
| 1 | 886 (70.3%) | 375 (29.7%) | 0.8069[f] |  |  |
| 2–3 | 15 (75.0%) | 5 (25.0%) |  |  |  |
| Gravidity |  |  |  |  |  |
| 1 | 66 (76.7%) | 20 (23.3%) | 0.4031 |  |  |
| 2–3 | 339 (70.0%) | 145 (30.0%) |  |  |  |
| ≥4 | 548 (69.8%) | 237 (30.2%) |  |  |  |
| Previous Live Births |  |  |  |  |  |
| 0 | 134 (74.4%) | 46 (25.6%) | 0.4385 |  |  |
| 1 | 195 (69.4%) | 86 (30.6%) |  |  |  |
| ≥2 | 590 (69.9%) | 254 (30.1%) |  |  |  |
| Surviving Children |  |  |  |  |  |
| 0 | 155 (72.1%) | 60 (27.9%) | 0.7587 |  |  |
| 1–2 | 478 (69.6%) | 209 (30.4%) |  |  |  |
| >2 | 284 (70.8%) | 117 (29.2%) |  |  |  |
| Previous Abortion |  |  |  |  |  |
| 0 | 529 (71.4%) | 212 (28.6%) | 0.2753 |  |  |
| ≥1 | 355 (68.5%) | 163 (31.5%) |  |  |  |
| Trimester at First Antenatal Care Visit |  |  |  |  |  |
| 1st (≤12 weeks) | 83 (79.0%) | 22 (21.0%) | 0.1409 |  |  |
| 2nd (13–26 weeks) | 652 (70.0%) | 279 (30.0%) |  |  |  |
| 3rd (≥27 weeks) | 278 (69.7%) | 121 (30.3%) |  |  |  |
| Total Antenatal Care Visits |  |  |  |  |  |
| 1–2 | 219 (70.9%) | 90 (29.1%) | 0.815 |  |  |
| 3–4 | 412 (71.7%) | 163 (28.3%) |  |  |  |
| >4 | 393 (69.9%) | 169 (30.1%) |  |  |  |
| **Delivery** |  |  |  |  |  |
| Year of Delivery |  |  |  |  |  |
| 2013–2014 | 339 (49.3%) | 349 (50.7%) | < .0001 | Ref | Ref |
| 2015–2017 | 685 (90.4%) | 73 (9.6%) |  | 0.10 (0.07–0.13) | <0.001 |
| Delivery Site |  |  |  |  |  |
| Jos University Teaching Hospital | 183 (69.6%) | 80 (30.4%) | 0.6024 |  |  |
| Other Clinic/Home/Road | 833 (71.2%) | 337 (28.8%) |  |  |  |
| Delivery Type |  |  |  |  |  |
| Vaginal/Assisted | 699 (68.3%) | 325 (31.7%) | 0.0276 |  |  |
| Emergency/Elective C-Section | 237 (74.8%) | 80 (25.2%) |  |  |  |
| Gestational Age at Delivery |  |  |  |  |  |
| Pre-term (<37 weeks) | 44 (73.3%) | 16 (26.7%) | 0.4943 |  |  |
| Full-term (≥37 weeks) | 875 (69.2%) | 390 (30.8%) |  |  |  |
| Infant Birthweight |  |  |  |  |  |
| Low (≤2.5 kg) | 233 (66.2%) | 119 (33.8%) | 0.0927 |  |  |
| Normal/High (>2.5 kg) | 688 (71.0%) | 281 (29.0%) |  |  |  |
| Infant Feed Method at Delivery |  |  |  |  |  |
| Exclusive Breast Feeding | 907 (71.6%) | 360 (28.4%) | 0.0611 |  |  |

(*Continued*)

**Table 4.** (Continued)

| | Viral Load Recorded | Viral Load Missing | chi-square test | Multiple Logistic Regression | |
|---|---|---|---|---|---|
| | Number (%) | Number (%) | p-value | aOR[a] (95% CI[b]) | p-value |
| Breast Milk Substitute Supplement | 95 (64.2%) | 53 (35.8%) | | | |

[a]aOR, adjusted odds ratio.

[b]CI, confidence interval.

[c]Denotes variables collected from the ART enrollment record, which was completed when the patient initiated ART in the APIN PEPFAR program. All other variables collected at antenatal booking or at delivery, as indicated.

[d]ART, antiretroviral therapy.

[e]ref, reference category.

[f]Fisher's exact test p-values reported when contingency table observations were less than or equal to five.

women with $\geq$95% MPR between months 18–24 postpartum. This postpartum decline again mimics the longitudinal trends of other sub-Saharan countries and identifies a crucial time period for intervention [10, 23, 24, 27].

The cumulative percentage of patients LTFU at 24 months postpartum in this study was 6.9%. This proportion is significantly lower than rates in other sub-Saharan countries such as Ethiopia and Malawi, where LTFU has ranged from 23%-24.5% [23, 24, 28]. The proportion LTFU in the general adult population with HIV in Nigeria has likewise been reported to be much higher, at 28% [29]. The focus on MTCT prevention at JUTH, through the APIN Public Health Initiatives, may have contributed to this improved retention among postpartum patients. Postpartum women with HIV may also be more motivated to continue ART (despite the difficulty in maintaining optimal adherence) during the first 24 months postpartum, while their infants are still being monitored for HIV infection. Exclusive breastfeeding was associated with viral suppression in bivariate analyses, suggesting a positive correlation with ART adherence, though not significant in the multiple regression, possibly due to collinearity with ART adherence which remained significant. Because our data were censored at 24 months, we could not assess outcomes afterward.

This study established risk factors for postpartum women becoming LTFU, previously unidentified in this patient population. Past studies have identified demographic determinants (i.e., younger age) and clinical determinants (i.e., viremia and missing CD4+ T-cell counts at delivery) as risk factors for pregnant women becoming LTFU after birth [30, 31]. Our study found the most significant risk factors for LTFU among postpartum patients were related to a patient's engagement and amount of contact time with the HIV clinic and antenatal care before delivery. Having a more recent HIV diagnosis, fewer antenatal care visits, and a delivery outside of JUTH increased the risk of LTFU. Higher gravidity also increased the risk of LTFU; women with prior pregnancies likely have children to care for at home, and less time to care for their own health. Strategies for retaining the postpartum population in HIV care should, therefore, identify and engage these high-risk patients with enhanced adherence counseling during pregnancy and in the first postpartum year. While costly, studies indicate that patient tracing and repeated home visits are successful methods for reconnecting with patients after becoming LTFU [32, 33].

Our study found 85.8% viral suppression at month 12 and 88.7% viral suppression at month 24 postpartum among those with viral load results. These numbers fall short of the 95% UNAIDS target for viral suppression. In comparison, South African studies found 14.7%–14.8% postpartum viral non-suppression [34, 35]. Importantly, in our study, of the 1497 ART-eligible, only 50.0% had viral load results at month 12, and 57.3% had viral load results at month 24. Viral load monitoring is a challenge in many resource-limited settings. A South

African study found only 12.6% of women had a viral load test by 9 months postpartum [34]. In our sensitivity analysis, patients missing postpartum viral load results were more likely to have shorter duration on ART and poorer adherence; therefore, our viral suppression rates in the postpartum population are likely overestimates. This may also explain why the proportion of patients with viral suppression did not decrease over the 24 months despite declining adherence. The 2018 Nigerian HIV/AIDS Indicator and Impact Survey found 77.1% viral suppression among adults on ART, which may be closer to what we might have observed in our study if viral load results were not missing [36].

Previous studies have identified younger age, shorter ART duration, and unsuppressed viral load at delivery as risk factors for unsuppressed viral load postpartum [12, 37, 38]. Our study similarly found unsuppressed viral load at delivery as a predictor of unsuppressed postpartum viral load; as unsuppressed viral load may indicate drug resistance, closer monitoring of these patients is needed. We additionally identified having poorer adherence, lower baseline CD4+ T-cell count, and more prior pregnancies as risk factors for unsuppressed postpartum viral load. Higher gravidity was a significant risk factor for both LTFU and unsuppressed viral load, and women with more children at home should be targeted for enhanced adherence counseling and supportive services.

While our study confirmed gaps in postpartum retention and adherence, solutions are not straightforward. Results of trials implementing phone calls or text message reminders to improve postpartum retention have been mixed [39–41]. Integrated care where mothers and infants are seen together at the PMTCT clinic postpartum has shown some promise. One study found 90% median ART adherence postpartum with 91% viral suppression in Uganda, where care was integrated, compared with 40% adherence postpartum with 57% viral suppression in South Africa, where women transferred to general ART services immediately after delivery [42]. A clinical trial found that integration of postpartum maternal and infant HIV care improved both retention and viral suppression at 12 months postpartum, but benefits did not continue after transfer to general ART services [43, 44]. Trials implementing 'mentor mothers' and community-based 'adherence clubs' have demonstrated improvements in viral suppression up to 24 months postpartum [45, 46].

While the large study population size and 24-month duration of individual postpartum follow-up strengthened this study, limitations remained. Our major limitation was missing data, with around 40% of viral load results missing at delivery and month 24, and 50% at month 12. Although viral load is the gold standard measure of ART effectiveness, it was not performed following the 12-monthly schedule. To accommodate for missing viral load data, missing baseline values were categorized separately and postpartum viral load time points were combined for the analyses, and a sensitivity analysis was performed to compare those with and without postpartum viral load results. Additionally, while MPR was used as a proxy measurement for ART adherence, without direct observation, it is possible that some patients picked up medication but did not adhere to their prescribed regimen. This study also exclusively focused on JUTH, a large urban tertiary hospital with an established HIV clinic since 2004, which may not be representative of Nigeria.

Finally, the impact of this study is limited by a lack of qualitative data. In addition to demographic and clinical barriers to ART retention and adherence like the ones we identified, significant individual (i.e., depression, understanding of ART), sociocultural (i.e., stigma, non-disclosure of HIV status), economic (i.e., financial resources, transportation), and structural (i.e., health worker attitudes) barriers persist for many people living with HIV in Nigeria and globally [25, 47–49]. Solutions to these long-standing barriers are elusive, as a Nigerian study that attempted a "continuous quality improvement" intervention found.[50] But so long as these barriers exist, any interventions may prove ultimately ineffective.

## Conclusions

As the risks for MTCT and adverse maternal health outcomes remain after birth, evaluation of retention, ART adherence, and viral suppression among postpartum mothers is critical. The cumulative percent of patients LTFU two years postpartum was lower for this Nigerian study population compared with postpartum patients in other sub-Saharan countries. As engagement with HIV and antenatal care decreases the risk of becoming LTFU, efforts to increase contact time among higher-risk patients should be initiated early in pregnancy. ART adherence among our postpartum population correlates with adherence rates in other sub-Saharan countries. The decline in adherence over the 24 months postpartum highlights the critical need for innovative adherence intervention strategies during this period. Viral suppression was considerably lower than the 95% UNAIDS target, and, importantly, the large percentage of missing viral load results was concerning. The causes of low viral load testing, whether policy-, funding-, and/or program-related must be addressed; point-of-care viral load monitoring should be considered.

For future studies, we recommend a longer follow-up time past 24 months to evaluate maternal retention, adherence, and viral load suppression after most infants have completed breastfeeding and HIV diagnostic testing. We also recommend including other urban and rural clinics in other regions for a more representative sample of postpartum women with HIV in Nigeria. Finally, we suggest surveys be administered throughout the MTCT prevention cascade to assess barriers to ART adherence and retention in care in this population as a start to understanding needs and considering interventions.

## Acknowledgments

The authors acknowledge with sincere gratitude the leadership, laboratory and data staff, clinicians, and patients of the APIN PEPFAR HIV program at the Jos University Teaching Hospital. We also thank the APIN Public Health Initiatives for providing the data used for the analyses.

## Author Contributions

**Conceptualization:** Clara M. Young, Charlotte A. Chang, Atiene S. Sagay, Olabanjo O. Ogunsola, Phyllis J. Kanki.

**Data curation:** Clara M. Young, Charlotte A. Chang.

**Formal analysis:** Clara M. Young, Charlotte A. Chang.

**Methodology:** Clara M. Young, Charlotte A. Chang, Godwin Imade, Phyllis J. Kanki.

**Supervision:** Atiene S. Sagay, Prosper Okonkwo, Phyllis J. Kanki.

**Visualization:** Clara M. Young, Charlotte A. Chang.

**Writing – original draft:** Clara M. Young, Charlotte A. Chang.

**Writing – review & editing:** Clara M. Young, Charlotte A. Chang, Atiene S. Sagay, Godwin Imade, Olabanjo O. Ogunsola, Prosper Okonkwo, Phyllis J. Kanki.

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
