## [Decision Letter · Decision Letter 0]

3 Jun 2024

PONE-D-24-14899Antiretroviral therapy retention, adherence, and clinical outcomes among postpartum women with HIV in NigeriaPLOS ONE

Dear Dr. Kanki,

Thank you for submitting your manuscript to PLOS ONE. After careful consideration, we feel that it has merit but does not fully meet PLOS ONE’s publication criteria as it currently stands. Therefore, we invite you to submit a revised version of the manuscript that addresses the points raised during the review process.

We look forward to receiving your revised manuscript.

Kind regards,

Hlengani Lawrence Chauke, PhD, MBCHB, BTh, Dip HIV Man, FCOG, MMED (O &G), MSc

Academic Editor

PLOS ONE

2. In this instance it seems there may be acceptable restrictions in place that prevent the public sharing of your minimal data. However, in line with our goal of ensuring long-term data availability to all interested researchers, PLOS’ Data Policy states that authors cannot be the sole named individuals responsible for ensuring data access (http://journals.plos.org/plosone/s/data-availability#loc-acceptable-data-sharing-methods).

Additional Editor Comments:

Thank you very much for submitting your study to PLOSONE. This is an important study that evaluated ARVs retention, adherence and viral suppression among postpartum women in Nigeria. You raise an important justification for conducting the study, i.e. that there are very few studies that focus on this important period of pregnancy. This is a retrospective study involving a review of data previously collected for the APIN Public Health Initiatives HIV program at JUTH, an initiative that was supported by PEPFAR. A total of 1535 women living with HIV were reviewed ,1497 of whom were on ART. This study not only provide information on ART retention, adherence and viral suppression but also important information on the risk factors for loss of follow up during this important period. Your conclusion is valid and on point. There is a need for prospective studies that focus on providing solutions on how to improve ART retention among postpartum women including qualitative studies to further enlighten us on this topic. The design of the study is appropriate for the research questions and objectives. The same applies regarding the choice of statistical measures. Overall, the manuscript is scientifically sound and well written. The following suggestions are recommended to further improve the paper:

1)Consistency with terminology: Please consider using women living with HIV (WLHIV) or HIV Positive women and not both

2) If possible, consider creating sub-categories under the <1000 bin (e.g. <200 and 200 – 1000) because of the differences in the definition of viral suppression between different regions/countries as well as the differences between Nigeria and US/WHO guidelines. This is important for the international readership and other researchers who might want to replicate the study in their settings. Kindly also respond to the additional comments (attached)  from the reviewers.

Thank you once more for sharing your work

Reviewers' comments:

Reviewer's Responses to Questions

**Comments to the Author**

1. Is the manuscript technically sound, and do the data support the conclusions?

Reviewer #1: Yes

Reviewer #2: Yes

2. Has the statistical analysis been performed appropriately and rigorously? 

Reviewer #1: Yes

Reviewer #2: Yes

3. Have the authors made all data underlying the findings in their manuscript fully available?

Reviewer #1: Yes

Reviewer #2: No

4. Is the manuscript presented in an intelligible fashion and written in standard English?

Reviewer #1: Yes

Reviewer #2: Yes

5. Review Comments to the Author

Reviewer #1: Thank you for the opportunity to review this well-written manuscript. It was a pleasure to read and highlights a very real problem in maternal and postpartum HIV care.

The authors conducted a retrospective review of data previously collected as part of the APIN Public Health

Initiatives HIV program at JUTH, supported by PEPFAR. In their study 1535 women living with HIV were reviewed and 1497 were on ART and thus eligible for further review in terms of viral suppression and loss to follow-up analysis. The authors showed that medication adherence (through medication possession ratio) declined from delivery through to 24 months postpartum and that close to 90% of women remained enrolled in care at 24 months post-delivery. Furthermore, the study adds to the literature by highlighting risk factors for loss to follow-up amongst postpartum women.

The authors raise a valid point in that further studies are needed to evaluate solutions for postpartum retention in care and the need for well-designed prospective studies to evaluate these variables. Furthermore, they highlight the need for qualitative studies in this field, something that is often shied away from.

I feel that the manuscript can be strengthened with two minor changes:

1) Please be consistent in your terminology – either “women living with HIV” or “HIV-positive women”

Page 3, Line 34: “ART coverage among pregnant and breastfeeding women with HIV…”

Page 3, Line 38: “among pregnant women with HIV…”

Page 4, line 59: “The study population included 1535 HIV-1-positive pregnant women”

2) In table 1, would it be possible to include a subcategory under the <1000 bin (e.g. <200 and 200 – 1000)? Nigeria seems to have a more liberal definition of viral suppression (1000cp/ml) compared to the US and WHO guidelines of 200cp/ml, so this may aid international readers in understanding the burden of HIV in Nigeria and this data could be compared to other international studies in the future

Thank you again for the opportunity to review your manuscript.

Reviewer #2: The manuscript is well written and makes an important contribution to the literature postpartum viral suppression in women living with HIV.

The are some minor questions that the authors need to address.

6. PLOS authors have the option to publish the peer review history of their article (what does this mean?). If published, this will include your full peer review and any attached files.

Reviewer #1: No

Reviewer #2: No

---

## [Author Response · Author response to Decision Letter 0]

2 Jul 2024

We have addressed each of the Editor and Reviewers comments in the 'Response to Reviewers' attachment and/or in the Manuscript, as appropriate. Our brief responses are below.

Journal Requirements:

1. We have made some minor tracked revisions to ensure that our manuscript meets PLOS ONE's style requirements to match the style templates.

2. In line with PLOS's Data Policy, we have added contact information for the APIN Public Health Initiatives Data Management Committee as a non-author institutional point of contact for data requests.

3. We have no changes to the Reference list.

Additional Editor Comments:

1. We have replaced prior terminology to 'women living with HIV' for consistency.

2. We have broken down the previous 'suppressed' baseline viral load category into subcategories (<200 cp/mL and 200-999 cp/mL) in Table 1, as suggested, due to regional and country differences in definition.

Reviewer 1:

1. We have replaced prior terminology to 'women living with HIV' for consistency.

2. We have broken down the previous 'suppressed' baseline viral load category into subcategories (<200 cp/mL and 200-999 cp/mL) in Table 1, as suggested, due to regional and country differences in definition.

Reviewer 2:

1. We have clarified the CD4 count monitoring schedule, which was every 6 months for all adult patients with HIV.

2. We have clarified the viral load monitoring schedule for adults newly initiating treatment, which was at 6 months, 12 months, and every 12 months thereafter, with an additional test at initiation for pregnant women.

3. We have clarified the subgroup for which mean viral load values were calculated, which was all patients that had any viral load recorded.

4. We have added the suggestion that breastfeeding may have contributed to ART adherence postpartum in the Discussion.

5. We have made a minor correction to the citation for reference #6, in which the journal name was displayed incorrectly.

Finally, we have also used the PACE digital diagnostic tool to check our figures and have re-uploaded the corrected files.

Again, we are thankful for all of the comments and suggestions which have helped to improve our manuscript. Kindly let us know if any of these or other items require further attention.

---

## [Editor Report · Decision Letter 1]

23 Jul 2024

Antiretroviral therapy retention, adherence, and clinical outcomes among postpartum women with HIV in Nigeria

PONE-D-24-14899R1

Dear Dr.  Kanki, Phylis, J

We’re pleased to inform you that your manuscript has been judged scientifically suitable for publication and will be formally accepted for publication once it meets all outstanding technical requirements.

Kind regards,

Hlengani Lawrence Chauke, MBCHB, BTh, Dip HIV Man, FCOG, MMED (O &G), MSc

Academic Editor

PLOS ONE

Additional Editor Comments (optional):

I would like to confirm that all the issues have been satisfactorily addressed. Thank you once again for all your effort.

---

## [Editor Report · Acceptance letter]

29 Jul 2024

PONE-D-24-14899R1 

PLOS ONE

Dear Dr. Kanki, 

I'm pleased to inform you that your manuscript has been deemed suitable for publication in PLOS ONE. Congratulations! Your manuscript is now being handed over to our production team.

Kind regards, 

on behalf of

Prof Hlengani Lawrence Chauke 

Academic Editor

PLOS ONE